# Methane and Carbon Dioxide Emission of Beef Heifers in Relation with Growth and Feed Efficiency

**DOI:** 10.3390/ani9121136

**Published:** 2019-12-12

**Authors:** Gilles Renand, Aurélie Vinet, Virginie Decruyenaere, David Maupetit, Dominique Dozias

**Affiliations:** 1UMR 1313 Génétique Animale et Biologie Intégrative, Université Paris-Saclay—Institut National de Recherche pour l’Agriculture, l’Alimentation et l’Environnement (INRAE)—AgroParisTech, Centre de Recherche de Jouy-en-Josas, 78350 Jouy-en-Josas, France; aurelie.vinet@inra.fr; 2Production and Sectors Department, Walloon Agricultural Research Centre, 8 rue de Liroux, 5030 Gembloux, Belgium; v.decruyenaere@cra.wallonie.be; 3UE 0332 Domaine Expérimental Bourges-La Sapinière, Institut National de Recherche pour l’Agriculture, l’Alimentation et l’Environnement (INRAE), Centre de recherche Val de Loire, 18390 Osmoy, France; david.maupetit@inra.fr; 4UE 0326 Domaine Expérimental du Pin, Institut National de Recherche pour l’Agriculture, l’Alimentation et l’Environnement (INRAE), Centre de recherche de Rennes, 61310 Le-Pin-au-Haras, France; dominique.dozias@inra.fr

**Keywords:** feed efficiency, methane, carbon dioxide, beef cattle

## Abstract

**Simple Summary:**

For sustainable meat production, beef farmers must make the best use of grass and roughage while limiting the carbon footprint of their herds. The genetic improvement in feed efficiency and enteric methane production of replacement heifers is possible if the recorded phenotypes are available. Intuitively, the relationship between the two traits should be negative, i.e., favorable, since the energy lost with the methane is not available for heifer metabolism. The measurement of feed efficiency requires several weeks of feed intake recording. The enteric methane emission rate can also be recorded over several weeks. The two traits of 326 beef heifers from two experimental farms were measured simultaneously for 8 to 12 weeks. The correlations between roughage intake, daily gain, and methane were all positive. The enteric methane emission rate was positively related to body weight, daily gain, and dry matter intake. The relationship with feed efficiency was slightly positive, i.e., unfavorable. Therefore, the two traits should be recorded simultaneously to evidence low-emitting and efficient heifers. This study also showed that replacing the feed intake recording with the carbon dioxide emission rate appeared potentially beneficial for selecting these low-emitting and efficient heifers.

**Abstract:**

Reducing enteric methane production and improving the feed efficiency of heifers on roughage diets are important selection objectives for sustainable beef production. The objective of the current study was to assess the relationship between different methane production and feed efficiency criteria of beef heifers fed ad libitum roughage diets. A total of 326 Charolais heifers aged 22 months were controlled in two farms and fed either a grass silage (*n* = 252) or a natural meadow hay (*n* = 74) diet. Methane (CH_4_) and carbon dioxide (CO_2_) emission rates (g/day) were measured with GreenFeed systems. The dry matter intake (DMI), average daily gain (ADG), CH_4_ and CO_2_ were measured over 8 to 12 weeks. Positive correlations were observed among body weight, DMI, ADG, CH_4_ and CO_2_. The residual feed intake (r_wg_DMI) was not related to CH_4_ or residual methane (r_wi_CH_4_). It was negatively correlated with methane yield (CH_4_/DMI): R_p_ = −0.87 and −0.83. Residual gain (r_w_iADG) and ADG/DMI were weakly and positively related to residual methane (r_wi_CH_4_): R_p_ = 0.21 on average. The ratio ADG/CO_2_ appeared to be a useful proxy of ADG/DMI (R_p_ = 0.64 and 0.97) and CH_4_/CO_2_ a proxy of methane yield (R_p_ = 0.24 and 0.33) for selecting low-emitting and efficient heifers.

## 1. Introduction

Beef producers face serious challenges in ensuring sustainable beef production. First, in many beef production systems, a large proportion of the feed is used by the cow breeding herd and forage feed self-sufficiency is a critical issue, especially with the added pressure from rising prices on cereal crops [1]. On the other hand, beef production contributes significantly to global greenhouse gas (GHG) emissions. In France, the carbon footprint comprises between 14.8 and 16.5 kg CO_2_/kg of live meat, depending on the production system [2], and 52% of this footprint is caused by enteric methane production [3].

Among the strategies to improve the utilization of forage resources and to reduce enteric methane production, the genetic selection of efficient and low-emitting animals is a promising option, since it allows for permanent and cumulative effects, as long as these traits are heritable. In fact, both traits have a moderate heritability. In a review of a large number of literature results with growing cattle, Berry and Crowley [4] calculated pooled heritability estimates of feed conversion ratio and residual feed intake (RFI): h^2^ = 0.33 and h^2^ = 0.23 respectively. There are very few estimates of the heritability of methane emission rate. In two studies with young cattle or lambs measured in respiratory chambers, the estimated heritability coefficients of methane emission rate and yield were slightly lower: h^2^ = 0.13 to h^2^ = 0.29 [5,6]. Therefore, genetic selection in order to improve feed efficiency and to reduce methane production is possible. While breeding programs to improve the feed efficiency of growing cattle have been in use in several beef breeds, there is still no breeding program for selecting low-emitting cattle. Different reasons exist for this; for example, the difficulty of measuring individual methane production and the uncertainty about the relationship between methane production and other characteristics. If a favorable correlation exists between feed efficiency and methane production, a mitigation of methane production could be obtained as a correlated response of the selection of more efficiently growing cattle. Based on first results [7,8], Waghorn and Hegarthy [9] and Basarab et al. [10] stated that selecting individuals with a low residual feed intake (RFI) would reduce methane production as a direct consequence of a reduction of feed intake. The results on the reduction of methane yield are more conflicting [11,12,13,14,15,16,17]. There is a need of new experiments to verify if low RFI beef cattle are actually low methane emitters before implementing breeding programs.

The first objective of the present study was to precisely estimate the phenotypic correlation between different feed efficiency criteria and methane production measures of beef heifers fed ad libitum on roughage-based diets. The second objective was to verify if the measure of carbon dioxide could be used as a proxy of feed intake, as suggested by Herd et al. [18], for ranking growing cattle on feed efficiency and methane yield.

## 2. Materials and Methods

### 2.1. Animals and Management

During this experiment, all animals were kept indoors and handled with care, following the INRAE ethics policy in accordance with the guidelines for animal research of the French Ministry of Agriculture. The approval number for ethical evaluation was APAFIS#14764-2018030610486896 v4.

The experiment was conducted in two experimental farms of the Institut National de Recherche pour l’Agriculture, l’Alimentation et l’Environnement (INRAE), Galle, near Bourges in the Centre Region and Borculo, near Le Pin-au-Haras in the Normandy Region. In both farms, purebred Charolais cows were inseminated with a unique set of 53 purebred Charolais bulls in order to have a common genetic background. After weaning, all the heifers were kept without any selection and without breeding until the experiment ended. They entered the testing barn at 22 months of age: 258 heifers, distributed in eight cohorts in the Galle farm and 75 heifers in two cohorts in the Borculo farm. After a 4-week adaptation period, the heifers were tested during a 12-week period in both farms. However, among the 10 cohorts, four cohorts had to be limited to 8 or 10 weeks due to a shortage in the availability of forages (2 cohorts) or due to technical problems, either with the feed weighing system (1 cohort) or with the GHG measurement system (1 cohort).

In the Galle farm, the heifers were housed in two or three pens (10–12 heifers per pen) equipped with individual troughs and electronically detected gates (American Calan Inc., Northwood, NH, USA). In the Borculo farm, the heifers were housed in two pens (16–21 heifers per pen) equipped with individual troughs and electronically detected gates (Proval, Ancenis, France). The heifers were allocated to pens according to age. Heifers in the same pen formed, therefore, a pen contemporary group (CG). There were 22 CG in 8 cohorts on the Galle farm and 4 CG in 2 cohorts on the Borculo farm. In both farms, a GreenFeed system (C-Lock Inc., Rapid City, SD, USA) was accessible in each pen. The floor was covered with wood shavings in Galle and with barley straw in Borculo farm. Heifers were fed ad libitum a roughage diet: grass silage (cultivated fescue) in Galle and natural meadow hay in Borculo. The heifers in Galle did not receive any concentrate complements while the heifers in Borculo received daily one-kilogram complements. The composition of the diets are reported in Table 1. The feed was poured into the troughs once per day shortly after 8:00 in the morning. The heifers were then blocked for 2 h before being freed for the rest of the day. Offered roughages were individually weighed daily. Feed refusals in each trough were removed and weighed three times weekly (Monday, Wednesday and Friday). The dry matter (DM) contents of offered roughages and feed refusals were then measured.

Faecal samples were taken directly from the rectum in order to estimate the chemical composition of the diet residues and the digestibility of the diet by near infrared reflectance spectroscopy (NIRS) analysis. Four times during the testing period, the heifers were sampled in the morning before feeding and the faeces was immediately oven dried for 72 h at 60 °C. The dried faeces were sent to the Walloon Agricultural Research Centre, Gembloux, Belgium where approximately 5 g of ground faecal samples (hammer mill—1 mm screen—Waterleau, BOA, Belgium) were submitted to NIRS scanning (NIRS system XDS, FOSS Electric, Hillerød, Denmark) and the absorption data recorded as log 1/R from 1100 to 2498 nm, every 2 nm (WINISI 1.5, FOSS Tecator Infrasoft International LCC, Hillerød, Denmark). The protein, NDF, ADF and ADL contents and the organic matter digestibility were estimated from the NIR spectra using the prediction equation developed by Decruyenaere et al. [19,20].

The methane and carbon dioxide emission rates were measured with the GreenFeed systems throughout the feed recording period (8 to 12 weeks). The GHG emissions were measured when the animals were visiting a concentrate feeder equipped with a head hood and an extractor fan for the capture of breath and eructation gases. At each visit, the gas emission rates were calculated, combining the gas concentrations (measured every second with a non-dispersive infrared analyzer) with airflow in the pipe (measured with a flow meter). The animals were attracted to the feeder with pellets that were distributed in small quantities. In both farms, the same type of pellets was delivered by the GreenFeed systems. Pellet delivery was programmed so that each heifer received 5 drops, i.e., 5 small quantities (36.3 ± 1.9 g/drop) per visit, with 45 s between each drop for a minimum measure duration of 3 min. The heifers were allowed to visit the GreenFeed systems a maximum of four times per day, separated by 6 h at least. An algorithm developed and applied by C-Lock Inc. calculated the GHG emission rates at each visit if the head of the animal was correctly positioned, as controlled by a laser beam.

### 2.2. Definition of Traits and Data Analyses

The definitions of all the recorded and calculated traits are presented in Table 2.

The animals were weighed twice at the start and at the end of the testing period and a single weight was recorded fortnightly. A regression of live weights on test day was performed for each animal using a Proc REG of the SAS statistical package [21]. The start test weight and the mid-test body weight (BW) were predicted when the test day was set to zero and to half of the test duration, respectively. A regression slope was used as a measure of average daily gain (ADG). Individual roughage dry matter (DM) intakes were calculated three times per week, combining the weight and DM content of offered roughages and feed refusals. The DM weight of the pellets delivered by the GreenFeed systems and the DM weight of the concentrate complement (Borculo farm) were added to the roughage DM intakes. The total DM intakes were averaged over the whole testing period to obtain a measure of daily dry matter intake (DMI). All the spot measures of gas emission rates of each heifer obtained over the whole testing period were averaged to obtain the daily methane emission rate (CH_4_) and daily carbon dioxide emission rate (CO_2_).

The feed efficiency was first calculated as the ratio ADG/DMI. Two other efficiency traits were computed as suggested by Koch et al. [22]. The first one, usually referred to as residual gain (RGain), was the difference between the observed daily gain and the expected daily gain, predicted from body weight and intake, calculated in each population as the residual (r_wi_ADG) of the following model: ADG = CG + α_1_MBW + α_2_DMI + r_wi_ADG, where CG is the fixed effect of the pen contemporary groups. The second efficiency trait was intake adjusted for differences in body weight and weight gain, usually referred to as residual feed intake (RFI). It was calculated in each population as the residual (r_wg_DMI) of the following model: DMI = CG + β_1_MBW + β_2_ADG + r_wg_DMI.

Similarly, two traits were defined for assessing differences in methane production among animals with different intakes and body weights. The methane yield was first calculated as the ratio CH_4_/DMI. A residual methane emission rate (r_wi_CH_4_) was then calculated as the residual of the following model: CH_4_ = CG + χ_1_MBW + χ_2_DMI + r_wi_CH_4_.

According to Herd et al. [18], who suggested that CO_2_ could be used as a proxy for feed intake for growing beef cattle fed ad libitum, five new feed and methane efficiency traits were calculated, with DMI replaced by CO_2_. The ratio of ADG to CO_2_ (ADG/CO_2_) was first calculated. A new residual gain (r_wc_ADG) was calculated as the residual of the following model: ADG = CG + δ_1_MBW + δ_2_CO_2_ + r_wc_ADG. A residual CO_2_ (r_wg_CO_2_) was calculated as the residual of the following model: CO_2_ = CG + ξ_1_MBW + ξ_2_ADG + r_wg_CO_2_. Similarly, two new traits were calculated for methane production among animals with different CO_2_ values. The ratio of CH_4_ to CO_2_ (CH_4_/CO_2_) was calculated and a new residual methane emission rate (r_wc_CH_4_) was calculated as the residual of the following model: CH_4_ = CG + φ_1_MBW + φ_2_CO_2_+ r_wc_CH_4_.

### 2.3. Data Analysis

The correlation coefficients among all the traits were calculated within the population after adjustment for the pen contemporary group (CG) effect, i.e., using the residuals of the following model: Y = CG + residual. The GLM and CORR procedures of the SAS statistical package were used for adjusting the observed traits and calculating the correlation coefficients between residuals.

In order to jointly quantify and compare the relationships of CH_4_ and CO_2_ with MBW, ADG and DMI in the two heifer populations, the following model was tested using Proc GLM of SAS: y = Farm + CG*Farm + γ_1_ MBW + γ_2_ ADG + γ_3_ DMI + γ_4_ MBW*Farm + γ_5_ ADG*Farm + γ_6_ DMI*Farm + error, where CG is the pen contemporary group.

## 3. Results and Discussions

Among the 333 heifers that entered the intake-recording barns, seven heifers had to be discarded from the analysis because they did not visit (*n* = 5) or irregularly visited (*n* = 2) the GreenFeed systems. The proportion of heifers that voluntarily visited the GreenFeed systems was very high. The observed individual variability was therefore fully representative of the variability that existed in these two Charolais herds. Eventually, the performance of 252 heifers in Galle and 74 heifers in Borculo could be used for the analysis of the individual differences in these two populations.

### 3.1. Recorded Traits

The descriptive statistics of the traits recorded in the two experimental farms are reported in Table 3. The heifers of both farms were very similar in age (22 months on average, CV ≤ 3%) and weight (500 kg on average, CV ≤ 10%) at the start of the experiment. The DMI of Galle heifers was 8.75 kg/d. The observed DM proportions of the diet ingredients were: 95% grass silage and 5% GreenFeed pellets. The DMI of Borculo heifers was 7.89 kg/d. The observed DM proportions of the diet ingredients were: 82% hay, 11% concentrates and 7% GreenFeed pellets. Although the DMI of Borculo heifers was only 10% lower than the Galle heifer DMI, their ADG was markedly lower: 360 vs. 932 g/d.

The organic matter digestibility difference was directly responsible for the observed efficiency difference: 72% for the Galle diet vs. 61% for the Borculo diet. The respective NIRS estimated chemical compositions of the Galle (grass silage) and Borculo (hay + 1 kg complement) diets were on average 135 vs. 107 (g/kg DM) for protein content, 590 vs. 675 for NDF, 313 vs. 339 for ADF and 106 vs. 78 for ADL. There were close relationships among the cohort averages of the diet chemical composition, diet digestibility and feed efficiency, as shown on Figure 1. Therefore, the differences in performance observed between the two farms were predominantly the consequence of differences in the digestibility of the roughage diets.

For gas emission rates, 3.3 spot measures per day were calculated on average by C-Lock for each heifer. The average visit duration was above 3 min in both farms; 3:39 and 4:01 min, respectively, in the Galle and Borculo farms. The hourly pattern of CH_4_ was very similar in the two farms (Figure 2). The emissions were at maximum (around 220 g/d) during the hours following the feed distribution and decreased regularly down to 160 g/d at the end of night before the next feed distribution.

During the 8–12-week testing period, over 200 spot measures per heifer were obtained on average for calculating the daily gas emission rates. This amount of spot measures was well above the minimum numbers, 20 to 50, recommended by several authors [23,24,25,26] for the precise measurement of CH_4_ and CO_2_ with GreenFeed systems. The CH_4_ averages were very similar in the two heifer populations: 205 g/d on average, while CO_2_ was lower for Borculo heifers compared to Galle heifers (6139 vs. 6918 g/d) as expected, due to their lower daily gain. The variability of the gas emission measures among heifers (CV = 7% and 10%) was in the same range as the body weight variability (CV = 9% and 10%) but lower than intake (CV = 11% and 15%) and gain (CV = 18% and 50%).

The phenotypic correlations among the recorded traits are reported in Table 4. As expected, all the correlation coefficients were positive. In the two populations, the correlation between gain and intake was only R_p_ = 0.37 and 0.28, i.e., in the lower range of phenotypic correlations (R_p_ = 0.42 on average) in experiments reviewed by Berry and Crowley [4]. In a recent publication where Charolais young bulls from the same Galle farm were fed pellet diet ad libitum from 9 to 17 months of age, the phenotypic correlation was R_p_ = 0.53 between DMI and ADG [27]. The lower coefficients of the current study could be explained by the diet energy content and the age of the heifers; at 22 months of age, the maintenance requirements represented approximately 34% of the gross energy intake while net energy of growth was less than 7% of GEI. The corresponding percentages were 26% and 11% for the finishing young bulls [27].

The correlation was noticeably high between CH_4_ and CO_2_ (R_p_ = 0.83 and 0.86). Similar high relationships, R_p_ = 0.78 on average (R_p_ = 0.57 to 0.91), were estimated with yearling beef cattle [16,18,26], dairy cows [28] or with lambs [29], measured either in respiratory chambers or with GreenFeed systems. The two emission rates were also highly correlated with the heifer body weight, slightly more for CO_2_ (R_p_ = 0.77 and 0.85) than for CH_4_ (R_p_ = 0.68 and 0.70). Similar results were obtained with dairy cows [30] and yearling beef cattle [14,31,32] measured with GreenFeed systems or with yearling beef cattle [5] and lambs [29] measured in respiratory chambers: R_p_ = 0.57 to 0.74 for the correlation between CH_4_ and BW and R_p_ = 0.71 to 0.87 for the correlation between CO_2_ and BW.

In ruminants, DMI and gross energy intake (GEI) are considered to be the predominant drivers of enteric methane production [33]. A meta-analysis of an international beef cattle database, used to determine a prediction equation for inventory purposes, actually showed that DMI was the most important predictor of CH_4_ [34]. The studies used in this meta-analysis covered a wide range of regions, systems or diets and showed a large variability of CH_4_ and DMI records: DMI = 8.13 ± 2.82 kg/d and CH_4_ = 161 ± 70.5 kg/d, much larger than the individual variability observed in the current study. When ranking animals for genetic selection, the variability in the recorded population should not depend on diet or environment differences but should primarily reflect the inter-animal variability, as in the current study. In the two populations of the current study, the correlation of CH_4_ and CO_2_ with DMI were only moderate (R_p_ = 0.36 to 0.52).

When the identification of low-emitting animals is required for genetic purposes, CH_4_ should be measured simultaneously to differences in feed intake with animals fed ad libitum. For precisely ranking growing cattle on feed intake, 35 to 45 recording days are recommended [35,36,37]. However, recording CH_4_ simultaneously to feed intake over several weeks is not possible with respiratory chambers or with the SF6 technic. It is possible with the GreenFeed system. To date, there has been a limited number of experiments with growing cattle simultaneously recorded for feed intake and gas emission using GreenFeed systems [14,16,18,26,31,32] with estimated correlation coefficients of R_p_ = 0.58 on average (R_p_ = 0.28 to 0.85) between CH_4_ and DMI and R_p_ = 0.65 on average (R_p_ = 0.40 to 0.89) between CO_2_ and DMI. The correlations of the current study were in the lower range of these published results. There is a clear need for further experimental results to obtain precise estimates of the relationship between CH_4_, CO_2_ and DMI among growing cattle of the same gender, age, physiology stage and fed ad libitum.

### 3.2. Calculated Growth and Methane Efficiency Traits

The statistics of growth and methane efficiency traits are reported in Table 5. The methane yield of Borculo heifers was 20% higher (26.4 vs. 24.0 g/kg) as compared to Galle heifers, while it was expected to be slightly lower (24.0 vs. 25.9 g/kg) according to the following equation published by Sauvant et al. [38] for ruminants fed roughage diets: CH_4_ (g/kg DM) = 0.137 OMD − 0.00009 OMD^2^ − 22.4 − 2.25 DMI/BW, where OMD is the organic matter digestibility (g/kg DM).

When calculating the residual methane emission (r_wi_CH_4_), the model explained R^2^ = 78% and 56% of CH_4_ differences in Galle and Borculo populations. The only comparable published results were obtained in a beef heifer and a steer population: R^2^ = 64% and R^2^ = 45% respectively [32].

The correlations among the recorded and calculated growth and methane efficiency traits are reported in Table 6. Due to the moderate R^2^ of the models (R^2^ < 33%) for the calculation of RGain (r_wi_ADG) and RFI (r_wg_DMI), these residuals were highly correlated with the two recorded traits: R_p_ = 0.89 and 0.96 for gain and R_p_ = 0.88 and 0.82 for intake. For methane production, the correlations between the recorded and residual trait were also high: R_p_ = 0.73 and 0.70.

Among the three feed efficiency criteria, the feed efficiency ratio (ADG/DMI) and the RGain (r_wi_ADG) were highly correlated (R_p_ = 0.81 and 0.99) and both traits were moderately and negatively correlated with RFI (r_wg_DMI): R_p_ = −0.37 on average (R_p_ = −0.67 to −0.20). In their review of phenotypic correlations, Berry and Crowley [4] also showed that RGain was more closely opposed to the feed conversion ratio (R_p_ = −0.71 on average) than to the RFI (R_p_ = −0.40 on average). Methane yield (CH_4_/DMI) and the residual methane emission (r_wi_CH_4_) were moderately correlated (R_p_ = 0.44 and 0.54). Herd et al. [18] found a closer relationship (R_p_ = 0.89) between CH_4_/DMI and r_i_CH_4_ with a limited number of animals.

Residual feed intake (r_wg_DMI) was independent of recorded and residual methane emissions but was markedly opposed to methane yield (R_p_ = −0.87 to −0.83). The other two feed efficiency traits, feed efficiency ratio and RGain, were weakly but positively correlated with CH_4_, methane yield and residual methane emission: R_p_ = 0.19 on average (R_p_ = 0.07 to 0.57). To date, there have been few studies with estimates of the phenotypic correlations between methane production and feed efficiency traits when both traits were recorded simultaneously. In a first experiment, the RFI of 76 steers was measured for 70 days and the CH_4_ for 10 days with SF6; no relationship was found between CH_4_ and RFI during the measurement period [8]. In another study, Fitzsimons et al. [12] measured the RFI of 20 beef heifers for 120 days and the CH_4_ with SF6 in two periods of 5 days; the RFI tended to have a positive correlation (R_p_ = 0.26) with CH_4_ and a negative correlation (R_p_ = −0.27) with CH_4_/DMI. Feed conversion efficiency, ADG/DMI, was not correlated with CH_4_ and CH_4_/DMI. Freetly and Brown-Brandl [39] measured the CH_4_ of 37 young steers and 46 heifers just after the RFI test period using head-hood calorimeters for 6 hours. No correlation was observed in both populations between RFI and CH_4_ adjusted for DMI and a weak positive correlation (R_p_ = 0.30 [*p* = 0.02]) was observed between ADG/DMI and CH_4_ adjusted for DMI in the heifer population. The results obtained by Herd et al. [14] in a study with 41 Angus steers and heifers fed ad libitum in a feedlot test, in which CH_4_ was measured with GreenFeed systems over 10 weeks, were in line with the current results: CH_4_, was not correlated with RFI nor with feed conversion. They found moderate negative correlations between methane yield and RFI (R_p_ = −0.54) and between methane yield and DMI/ADG (R_p_ = −0.47). They also found weak negative correlations between residual methane emission adjusted for DMI and RFI (R_p_ = −0.29 [*p* < 0.10]) and between residual methane emission and DMI/ADG (R_p_ = −0.22 [*p* > 0.10]). The phenotypic correlations estimated in these five experimental populations, in addition to the current results, have shown that the relationship between feed efficiency and methane production traits are tenuous and may depend on the population and diet. Overall, these results show that in growing cattle fed ad libitum, (i) CH_4_ is undoubtedly, but moderately, associated with intake; (ii) CH_4_ is predominantly independent of residual feed intake and feed conversion; (iii) there is a slight tendency for the methane yield to be negatively correlated with residual feed intake and feed conversion.

Another approach was to select animals with extreme RFI values and then measure methane production. In the pioneer study of Nkrumah et al. [7], eight low-RFI and 11 high-RFI steers selected out of 306 steers were placed in metabolism crates, then CH_4_ was measured for two 16-h periods with head-hood calorimetry systems. With feed limited to a fixed allowance, the low-RFI steers produced 25% less methane than high-RFI steers [*p* = 0.04]. In the study of Alemu et al. [16], eight low-RFI and eight high-RFI heifers selected out of 98 heifers were measured with GreenFeed systems in a pen over two 25-d periods (fed ad libitum) and over 2 days in respiratory chambers (20% less DMI). When recorded with the GreenFeed systems, the low-RFI heifers produced 9% less methane than high-RFI heifers [*p* = 0.02]. No difference was observed when measured in the respiratory chambers [*p* = 0.40]. The methane yield was also not different in group pens or in respiratory chambers [*p* = 0.25 and 0.99]. In the other studies designed to measure the response to a selection of RFIs, there was no significant CH_4_ difference between low- and high-RFI cattle. In a first study, Mercadante et al. [13] measured CH_4_ of 22 low-RFI and 24 high-RFI animals selected among 118 male and female yearling Nellore cattle: CH_4_ was measured with the SF6 technic and was not different between the two groups: 142 and 144 g/day, respectively [*p* = 0.69]. However, the methane yield was significantly higher [*p* < 0.001] in the low-RFI (25.1 g/kg DMI) compared to the high-RFI (22.8 g/kg DMI) group. In a second study with Nellore cattle, Oliveira et al. [17] observed no difference in the CH_4_ measured with the SF6 technic between 25 low-RFI and 22 high-RFI animals selected among 159 male and female yearling Nellore animals when tested in a feedlot then at pasture. Methane yield, when measured in feedlot, was not different between low- and high-RFI animals. McDonnell et al. [15] selected 14 low-RFI and 14 high-RFI heifers among 86 yearling beef heifers and measured CH_4_ with the SF6 technic in three successive periods with three different conditions and diets. There was no diet x RFI group interaction and the CH_4_ difference between low- and high-RFI groups (156 and 146 g/d respectively) was not significant [*p* = 0.11]. However, the low-RFI heifers had a greater methane yield than the high-RFI heifers: 22.4 vs. 20.2 g/kg DM, respectively [*p* = 0.034], whatever the diet. In a study with dairy Holstein and Jersey cattle, 14 low-RFI and 14 high-RFI heifers were selected among 140 heifers within each breed and then measured with GreenFeed during 18 or 25 days [40]. No RFI group difference was observed for CH_4_ [*p* = 0.60], but for methane yield, the low-RFI heifers produced significantly [*p* < 0.01] (10%) more methane per kg DM (22.7 g/kg DM) than high-RFI heifers (20.7 g/kg DM). With the exception of the first study, where a large difference was observed between low- and high-RFI animals, the other RFI selection studies confirmed the weak relationship between RFI and CH_4_. In addition, three of these studies showed a positive increase in methane yield when selecting low-RFI as compared to high-RFI animals.

Given the varied and sometimes contradictory results of the literature, large recorded populations are needed to highlight the weak relationship that may exist between feed efficiency and enteric methane production. Thus, the current experience, with 326 recorded heifers, has allowed for a precise estimate of the relationships between CH_4_, feed intake and growth. In a first step, the simple regressions of CH_4_ and CO_2_ on MBW, ADG and ADG were calculated. There was no difference between the two heifer populations for any of the regression slopes on weight, gain or intake. All the simple regression, calculated on the within population pooled data, were highly significant. The slope of CH_4_ on DMI (7.5 ± 1.3 g/d/kg) was within the range of values estimated by Bird-Gardiner et al. [33] and lower than the 20.1 g/d/kg estimated by Manafiazar et al. [26]. The slope of CO_2_ on DMI (202 g/d/kg), was lower than the slopes estimated by Arthur et al. [31] and Manafiazar et al. [26].

In Table 7 are reported the results of the joint analysis of variance of the whole data, including the multiple regressions on MBW, ADG and DMI.

When the GHG emission rates were simultaneously regressed on MBW, ADG and DMI, only the first two traits remained related to CH_4_ and CO_2_, while partial regressions on DMI were no longer significant. In these two populations, the positive relationship between CH_4_ and DMI was generated by the correlations of both traits with body weight. The final models were: CH_4_ (g/d) = 7.2 ± 11.8 + 1.6 ± 0.1 MBW (kg) + 0.021 ± 0.005 ADG (g/d) + 0.8 ± 0.7 DMI (kg/d), R^2^ = 0.77; CO_2_ (g/d) = 793 ± 274 + 49 ± 3 MBW (kg) + 0.51 ± 0.12 ADG (g/d) + 15 ± 17 DMI (kg/d), R^2^ = 0.84.

These equations show that among heifers with the same weight and the same intake, the heifers with higher daily gain, i.e., which were more efficient, produced more CH_4_ and CO_2_. This gas production was relatively modest however: heifers with daily gain one SD above the mean produced 1.7% more methane and 1.3% more carbon dioxide. The energy of eructed methane represented approximately 8% of the gross energy intake (GEI) of these heifers. Although this lost energy could not be used for heifer metabolism, it was probably compensated for by the increased metabolizable energy available for growth.

Both the methane and the metabolizable energy of the diet depend on the digestion process in the rumen—an increased digestion can provide more nutrients for the animal metabolism, as well as increased enteric methane production when roughage diets are consumed. In the review of Richardson and Herd [41], digestibility was assumed to account for 10% of the variation in RFI on average in beef cattle, while Lovendahl et al. [42], in their review of dairy cattle, concluded that between-cow variation in digestibility alone was too small to explain the observed variation in feed efficiency. Nonetheless, higher digestibility of DM or NDF was found in low-RFI as compared to high-RFI cattle in the studies of McDonnell et al. [15] and Oliveira et al. [17] with beef cattle and Olijhoek et al. [43] with dairy Holstein cows. On the other hand, no significant difference was observed in the studies of Nkrumah et al. [7], Fitzsimons et al. [12] and Fitzsimons et al. [44] with beef cattle and Olijhoek et al. [43] with dairy Jersey cows and Fischer et al. [45] with dairy Holstein cows. Interestingly, a higher digestibility and a higher methane yield were observed in low-RFI beef heifers tested with three different diets by McDonnell et al. [15] and in low-RFI beef Nellore tested in feedlot by Oliveira et al. [17].

The mechanisms causing among-animal variations in digestibility and enteric methane production are still unclear. The mean retention time (MRT) of the digesta in the rumen is one of the possible mechanisms. Huhtanen et al. [46] developed a mechanistic model to evaluate among-animal variations in MRT on organic matter digestibility and methane emissions. This model was developed for dairy cows and sheep, but the results can be extended to any other ruminant fed a total mixed ration or a roughage diet. They predicted that higher digestibility and higher methane emission were associated with an increased retention time. Such relationships between MRT and CH_4_ or methane yield were observed in sheep [47,48,49]. The composition of the rumen microbiome is the core factor related to digestion and enteric methane production and consequently, to efficiency. Recent sequencing possibilities of the metagenome or at least the 16S and 18S rRNA genes have broadened our knowledge on the extremely variable and abundant microbial communities in the rumen fluid. Analyses of the microbiome have been conducted in beef and dairy cattle for the characterization of the communities linked to methane emission or feed efficiency [50,51,52,53,54]. In the first four studies, the abundance of a number of archea and bacteria communities was actually identified in higher emitters.

Whatever the biology mechanisms involved in methane production and feed efficiency, the results of the current study confirm the tenuous and possibly unfavorable relationship between these two essential traits. With the assumption that the genetic correlation would be of similar magnitude, no correlated reduction of enteric methane production would be expected when selecting efficient beef heifers. Therefore, implementing a reference population for the genetic improvement of both traits requires the simultaneous phenotyping of growth, feed intake and methane production in growing cattle.

### 3.3. Calculated Growth and Methane Efficiency Traits Adjusted for Carbon Dioxide Emission

The statistics of the traits adjusted for CO_2_ are reported in Table 8.

The correlations among the recorded and calculated growth and methane efficiency traits are reported in Table 9.

The two new growth efficiency criteria, ADG/CO_2_ and r_wc_ADG, were strongly correlated with ADG (R_p_ = 0.88 to 0.99) and weakly with DMI (R_p_ = 0.19 to 0.22). Consequently, they were also highly correlated with ADG/DMI and residual gain (r_wi_ADG): R_p_ = 0.87 on average (R_p_ = 0.64 to 0.97). Arthur et al. [31] obtained a correlation of a similar amplitude, R_p_ = 0.73, between CO_2_/ADG and DMI/ADG with 326 steers measured with GreenFeed. The results of the present study with beef heifers fed ad libitum with roughage diets confirm that adjusting growth performance with CO_2_ could be used as a proxy for improving growth efficiency in the absence of DMI measurements. On the other hand, the residual of CO_2_ from MBW and ADG (r_wg_CO_2_) was not related to RFI (R_p_ = 0.04 and 0.10), whereas Arthur et al. [31] found a low and significant correlation: R_p_ = 0.27. The correlations of the CH_4_/CO_2_ ratio with CH_4_ were R_p_ = 0.58 and 0.61, the same values obtained by Herd et al. [18] with yearling beef animals measured with GreenFeed (R^2^ = 0.40) and Jonker et al. [29] with lambs measured in respiratory chambers, R_p_ = 0.65. The correlations of CH_4_/CO_2_ with CH_4_/DMI were low, R_p_ = 0.24 and 0.33, much lower than the estimates obtained by Herd et al. [18] (R^2^ = 0.49) and Jonker et al. [29], R_p_ = 0.84. The residual methane emission from MBW and CO_2_ (r_wc_CH_4_) was correlated with CH_4_: R_p_ = 0.51 and 0.56, when the R^2^ of the relationship estimated by Herd et al. [18] was R^2^ = 0.61. Strong correlations, R_p_ = 0.70 and 0.79, were observed between the residuals of CH_4_ from MBW and CO_2_ or DMI. The R^2^ of the relationship estimated by Herd et al. [18] was R^2^ = 0.68. In the absence of DMI recording, it would be profitable to use the residuals of ADG and CH_4_ from MBW and CO_2_ (r_wc_ADG and r_wc_CH_4_) as proxies for ranking animals on residual gain and residual methane emission. More studies are needed to evaluate these relationships in growing cattle. Eventually, it is worth noting that when using CO_2_ in place of DMI, (i) the growth efficiency proxies (ADG/CO_2_ and r_wc_ADG) were independent of methane yield (CH_4_/DMI) and r_wic_CH_4_; (ii) the methane efficiency proxies (CH_4_/CO_2_ and r_wc_CH_4_) were independent of growth efficiency traits.

## 4. Conclusions

The main result of this study was the lack of a relationship between CH_4_ and RFI. It confirms most of the results of the literature. A slightly positive, i.e., unfavorable, relationship was observed, however, when the feed efficiency was appreciated by the residual daily gain or the gain to intake ratio. No mitigation in enteric methane production is expected, therefore, when selecting efficient growing cattle. Consequently, the implementation of a breeding program for a better utilization of roughage-based diets simultaneously with a reduction of enteric methane production requires the recording of both trait phenotypes.

The second result is the potential benefit of using the GreenFeed CO_2_ measurements in calculating proxies for the selection of feed efficiency and methane emission of growing cattle when no DMI measurement is available. Further studies are needed however to confirm the phenotypic relationships observed in this study.

The phenotypic correlations estimated in this study must be supplemented by estimates of the genetic parameters. Much larger reference populations with the measurement of these phenotypes are needed to quantify the genetic progress that can be achieved for the different feed efficiency and methane production traits. In addition, more detailed physiology and nutrition studies will be useful to understand the different mechanisms controlling feed intake, growth, and methane and carbon dioxide emission of growing cattle.

## Figures and Tables

**Figure 1 animals-09-01136-f001:**
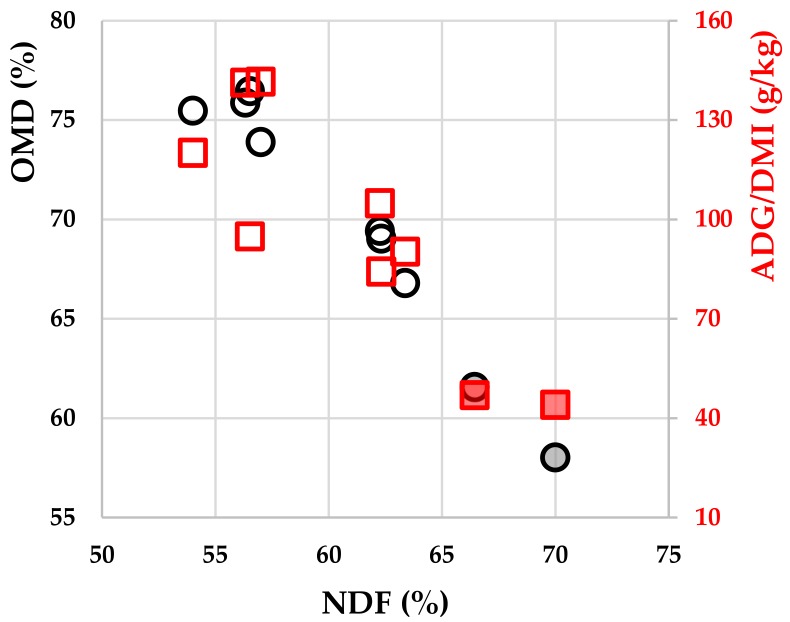
Cohort averages of estimated organic matter digestibility (OMD) (black circles) and feed efficiency ratio (ADG/DMI) (red squares) vs. NDF of Galle heifers (empty) and Borculo heifers (colored filling).

**Figure 2 animals-09-01136-f002:**
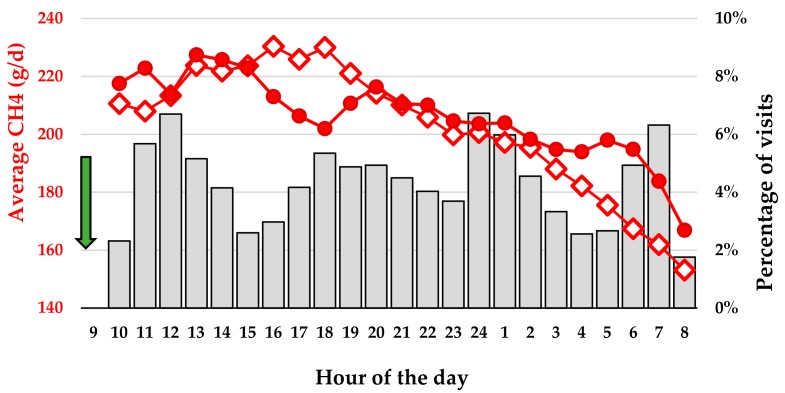
Average number of visits to the GreenFeed systems for all heifers (bar graph) and average CH_4_ emission rate for Galle heifers (empty) and Borculo heifers (colored filling) over 23 h starting at 9:00 when the heifers were freed. The troughs were filled with roughage (green arrow) at 8:00.

**Table 1 animals-09-01136-t001:** Diet characteristics.

Ingredients	Grass Silage ^†^	Hay	Concentrate Complement *	GreenFeed Pellet **
Dry matter (g/kg)	264 [210–352]	864	876	889
Chemical composition (g/kg DM)				
Ash	105 [90–117]	82	60	89
Crude Protein	111 [107–118]	78	162	153
Cellulose	295 [284–307]	317	53	171
Net energy (MJ/kg DM)	6.04 [5.96–6.19]	3.90	8.03	5.96

^†^ Overall mean and [cohort extremes]; * Concentrate complement: extracted soybean meal, corn, wheat bran, sugar cane molasses; ** GreenFeed pellet: wheat middling and bran, dehydrated alfalfa hay and dehydrated beat pulp.

**Table 2 animals-09-01136-t002:** Definition of traits.

Trait	Unit	Abbreviation	Formula
Mid-test body weight	kg	BW	Predicted live weight by regression at mid-test
Metabolic body weight	Kg^0.75^	MBW	BW^0.75^
Dry matter intake	kg/d	DMI	Daily average of roughage plus pellet dry matter intake
Average daily gain	g/d	ADG	Regression coefficient of body weights on time
Methane emission rate	g/d	CH_4_	Daily average of methane emission spot measures
Carbon dioxide emission rate	g/d	CO_2_	Daily average of carbon dioxide emission spot measures
Feed efficiency ratio	g/kg	ADG/DMI	ADG divided by DMI
Residual gain from MBW and DMI	g/d	r_wi_ADG	ADG—expADG where expADG is obtained by the regression of ADG on MBW and DMI
Residual intake from MBW and ADG	kg/d	r_wg_DMI	DMI—expDMI where expDMI is obtained by the regression of DMI on MBW and ADG
Methane yield	g/kg	CH_4_/DMI	CH_4_ divided by DMI
Residual methane from MBW and DMI	g/d	r_wi_CH_4_	CH_4_—expCH_4_ where expCH_4_ is obtained by the regression of CH_4_ on MBW and DMI
Gain to carbon dioxide ratio	g/kg	ADG/CO_2_	ADG divided by CO_2_
Residual gain from MBW and CO_2_	g/d	r_wc_ADG	ADG—expADG where expADG is obtained by the regression of ADG on MBW and CO_2_
Residual carbon dioxide from MBW and ADG	g/d	r_wg_CO_2_	CO_2_—expCO_2_ where expCO_2_ is obtained by the regression of CO_2_ on MBW and ADG
Methane to carbon dioxide ratio	g/kg	CH_4_/CO_2_	CH_4_ divided by CO_2_
Residual methane from MBW and CO_2_	g/d	r_wc_CH_4_	CH_4_—expCH_4_ where expCH_4_ is obtained by the regression of CH_4_ on MBW and CO_2_

**Table 3 animals-09-01136-t003:** Statistics of traits recorded in the two experimental farms.

Trait ^‡^	Unit	Galle Heifers (*n* = 252)			Borculo Heifers (*n* = 74)		
		Mean	SD *	CV †	Mean	SD *	CV †
Start age	d	678	6	1%	665	21	3%
Start weight	kg	499	44	9%	495	49	10%
Organic matter digestibility		0.723	0.017	2%	0.605	0.015	2%
BW	kg	534	46	9%	508	49	10%
DMI	kg/d	8.75	1.31	15%	7.89	0.87	11%
ADG	g/d	932	163	18%	360	180	50%
Spot measures per day		3.4	0.9	26%	3.2	0.5	15%
Total number of measures		220	59	27%	232	39	17%
Visit duration	min:s	3:39	0:12	6%	4:01	0:19	8%
CH_4_	g/d	205	20	10%	206	17	8%
CO_2_	g/d	6918	544	8%	6139	411	7%

^‡^ See Table 1 for description of traits; * standard deviation of traits adjusted for contemporary group effects; † CV of residual traits = SD * divided by the non-adjusted variable mean.

**Table 4 animals-09-01136-t004:** Correlations among traits recorded in the two experimental populations *.

Trait ^‡^	DMI	ADG	CH_4_	CO_2_
BW	0.42 ^a^	0.40 ^a^	0.68 ^a^	0.77 ^a^
	0.53 ^a^	0.11	0.70 ^a^	0.85 ^a^
DMI		0.37 ^a^	0.36 ^a^	0.38 ^a^
		0.28 ^c^	0.48 ^a^	0.52 ^a^
ADG			0.44 ^a^	0.46 ^a^
			0.26 ^c^	0.21
CH_4_				0.86 ^a^
				0.83 ^a^

* Galle heifers on first line, Borculo heifers on second line; ^‡^ See Table 1 for description of traits; ^a^ Correlation coefficient significantly different from zero at *p* < 0.001; ^b^ Correlation coefficient significantly different from zero at *p* < 0.01; ^c^ Correlation coefficient significantly different from zero at *p* < 0.05.

**Table 5 animals-09-01136-t005:** Statistics of calculated growth and methane efficiency traits.

Trait ^‡^	Unit	Galle Heifers (*n* = 252)			Borculo Heifers (*n* = 74)		
		Mean	SD *	CV ^†^	Mean	SD *	CV †
ADG/DMI	g/kg	109	22	20%	46	22	49%
r_wi_ADG	g/d	0	145	16%	0	173	48%
r_wg_DMI	kg/d	0	1.15	13%	0	0.71	9%
CH_4_/DMI	g/kg	24.0	3.8	16%	26.4	2.9	11%
r_wi_CH_4_	g/d	0	14	7%	0	12	6%

^‡^ See Table 1 for description of traits; * standard deviation of traits adjusted for contemporary group effects; ^†^ CV of residual traits = SD * divided by the non-adjusted variable mean.

**Table 6 animals-09-01136-t006:** Correlations among calculated growth and methane efficiency traits *.

Trait ^‡^	ADG/DMI	r_wi_ADG	r_wg_DMI	CH_4_/DMI	r_wi_CH_4_
BW	0.03	0.00	0.00	0.02	0.00
	−0.02	0.00	0.00	0.04	0.00
DMI	−0.43 ^a^	0.00	0.88 ^a^	−0.77 ^a^	0.00
	0.04	0.00	0.82 ^a^	−0.67 ^a^	0.00
ADG	0.63 ^a^	0.89 ^a^	0.00	−0.08	0.20 ^c^
	0.97 ^a^	0.96 ^a^	0.00	−0.08	0.21
CH_4_	0.14 ^c^	0.17 ^b^	0.04	0.27 ^a^	0.73 ^a^
	0.13	0.15	0.08	0.30 ^b^	0.70 ^a^
CO_2_	0.14 ^c^	0.16 ^c^	0.02	0.17 ^c^	0.46 ^a^
	0.08	0.10	0.05	0.15	0.31 ^c^
ADG/DMI		0.81 ^a^	−0.67 ^a^	0.57 ^a^	0.21 ^a^
		0.99 ^a^	−0.20	0.07	0.19
r_wi_ADG			−0.23 ^a^	0.10	0.23 ^a^
			−0.26 ^c^	0.13	0.22
r_wg_DMI				−0.87 ^a^	−0.05
				−0.83 ^a^	−0.06
CH_4_/DMI					0.44 ^a^
					0.54 ^a^

* Galle heifers on first line, Borculo heifers on the second line; ^‡^ See Table 1 for description of traits; ^a^ Correlation coefficient significantly different from zero at *p* < 0.001; ^b^ Correlation coefficient significantly different from zero at *p* < 0.01; ^c^ Correlation coefficient significantly different from zero at *p* < 0.05.

**Table 7 animals-09-01136-t007:** Joint analysis of variance for CH_4_ and CO_2_ of heifers of the two farms *.

Trait	ddl	CH_4_		CO_2_	
Effect		F Value	Pr > F	F Value	Pr > F
Farm	1	1.8	0.19	1.5	0.23
Contemporary Group*Farm	24	19.6	<0.001	22.2	<0.001
MBW	1	108.0	<0.001	196.7	<0.001
ADG	1	10.8	0.001	9.6	0.002
DMI	1	1.2	0.28	0.5	0.48
MBW*Farm	1	0.6	0.44	1.9	0.16
ADG*Farm	1	0.4	0.52	1.7	0.19
DMI*Farm	1	0.3	0.56	0.1	0.75

* model: CH_4_ = Farm + CG*Farm + γ_1_MBW + γ_2_ADG + γ_3_DMI + γ_4_MBW*Farm + γ_5_ADG*Farm + γ_6_DMI*Farm + error.

**Table 8 animals-09-01136-t008:** Statistics of calculated growth and methane traits adjusted for carbon dioxide.

Trait	Unit	Galle Heifers (*n* = 252)			Borculo Heifers (*n* = 74)		
		Mean	SD *	CV ^†^	Mean	SD *	CV ^†^
ADG/CO_2_	g/kg	136	21	15%	59	28	48%
r_wc_ADG	g/d	0	144	15%	0	174	48%
r_wg_CO_2_	kg/d	0	337	5%	0	212	3%
CH_4_/CO_2_	g/kg	29.8	1.5	5%	33.5	1.6	5%
r_wc_CH_4_	g/d	0	10	5%	0	10	5%

* standard deviation of traits adjusted for contemporary group effects; ^†^ CV of residual traits = SD* divided by the non-adjusted variable mean.

**Table 9 animals-09-01136-t009:** Correlations among calculated growth and methane traits adjusted for carbon dioxide *.

Trait ^‡^	ADG/CO_2_	r_wc_ADG	r_wg_CO_2_	CH_4_/CO_2_	r_wc_CH_4_
BW	0.09	0.00	0.00	0.13 ^c^	0.00
	0.00	0.00	0.00	0.07	0.00
DMI	0.22 ^a^	0.20 ^b^	0.03	0.10	0.05
	0.20	0.19	0.08	0.12	0.09
ADG	0.89 ^a^	0.88 ^a^	0.00	0.14 ^c^	0.08
	0.99 ^a^	0.97 ^a^	0.00	0.16	0.15
CH_4_	0.08	0.04	0.49 ^a^	0.58 ^a^	0.51 ^a^
	0.14	0.09	0.40 ^a^	0.61 ^a^	0.56 ^a^
CO_2_	0.03	0.00	0.62 ^a^	0.09	0.00
	0.07	0.00	0.52 ^a^	0.06	0.00
ADG/DMI	0.64 ^a^	0.65 ^a^	0.01	0.05	0.04
	0.97 ^a^	0.96 ^a^	−0.03	0.11	0.11
r_wi_ADG	0.91 ^a^	0.94 ^a^	−0.01	0.09	0.07
	0.97 ^a^	0.95 ^a^	−0.03	0.13	0.12
r_wg_DMI	−0.02	−0.01	0.04	0.03	0.03
	−0.02	−0.02	0.10	0.06	0.08
CH_4_/DMI	−0.17 ^b^	−0.17 ^b^	0.28 ^a^	0.24 ^a^	0.25 ^a^
	−0.10	−0.14	0.24 ^c^	0.33 ^b^	0.32 ^b^
r_wi_CH_4_	0.01	0.04	0.68 ^a^	0.67 ^a^	0.70 ^a^
	0.17	0.08	0.56 ^a^	0.79 ^a^	0.79 ^a^

* Galle heifers on first line, Borculo heifers on second line; ^a^ Correlation coefficient significantly different from zero at *p* < 0.001; ^b^ Correlation coefficient significantly different from zero at *p* < 0.01; ^c^ Correlation coefficient significantly different from zero at *p* < 0.05.

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
