# Peer review of "Methane and Carbon Dioxide Emission of Beef Heifers in Relation with Growth and Feed Efficiency"

_animals, 2019, doi:10.3390/ani9121136_

Round 1

Reviewer 1 Report

The manuscript has been improved but the imbalance between the two sites and some doubts about statistics are still there. Again the main concern is that this study can say something about selection and how to select low emitting animals.

Reviewer 2 Report

The manuscript has been revised following the suggestions of the reviewer, and it is more clear.

Other comments

L94: Use GHG instead of greenhouse gas after describing it for the first time.

L127: Please use CH4 or methane and carbon dioxide or CO2 uniformly, not both throughout the manuscript.

Table 3: days to d under unit

L395: the prediction equations are confusing when considered other prediction equations. Authors can use simple prediction equations followed multiple regression equations for better understanding. They should also discuss it with respect to other literature.

Reviewer 3 Report

The manuscript was corrected by the authors, but in my opinion there were no radical changes improving the understanding of the text for the reader. The manuscript is still too long, especially the results and discussion section.
In an earlier review of this work, I gave only examples where the authors describe the results of research by other authors. I understand that some information are necessary and must be included in the discussion, however this part cannot be like a review of the literature. Of course, the authors decide what should be included in the manuscript.
My task is substantive and general assessment of the manuscript, improving its quality. In general, the manuscript is of good quality, which I wrote in an earlier review, but the presentation of the research and its discussion has still not been sufficiently corrected.
However, the authors decide on the appearance of their manuscript.
I do not notice substantive errors.

Reviewer 4 Report

The author addressed the reviewer suggestions. The paper is ready to be published in present form. 

This manuscript is a resubmission of an earlier submission. The following is a list of the peer review reports and author responses from that submission.

Round 1

Reviewer 1 Report

The manuscript describe an experiment that has been designed for measuring methane and carbon dioxide production in two experimental farms in which different diet has been feed to heifers. Although the topic is timely and extremely interesting the experimental design is confused as well as the statistics adopted. The main concern is about the quantity and quality of the data collected that make ay conclusion especially from a genetic point of view very speculative

Title do not reflect the content. For instance: genetic selection …..Assuming that it was a goal of the study is missed even if this topic in in the second line of both interpretative abstract and the regular one.

L15-28 please check for the English (all the paper)

L26Please explicit “unfavourable” for what

L28 is genetic selection addressed in the paper?

L37-39 not clear please rephrase

L41 “ what do authors mean with “was interesting”

L54 -67 is focused on breeding and genetics heritability of selected traits never really addressed in the paper.

L62 are these heritability values (h. = 0.13 to h. = 0.29) acceptable for start a selection program for them?

L73-81 authors mixed up aim, methods and introduction

L96-101 not clear how many cohort where used finally. Furthmeore there was a big ambalance between th etwo farm and therefore expereimenatli groups.

L102 115 please clarify better, also with scheme, how many pens were used (3 pens x 12 haifer=36 higfers bit less than 268…). Confuse and vague.

L107-109 please provide more details about the diets. What is “grass silage (cultivated fescue)”

L114-124 all these information declined according to the farm must be reported in a table with diet ingredients and nutritional fact of th complete diets.

Heifers were Charolaise why UFL???? Now 8 cohorts?

L141-142 move here what is reported in line 147-151. However 36 g are probably very few pellet…..were they perceived by the animals

L141-146 has been validated or reported in another peer review article?

L161 was it an INDIVIDUAL DMI???

L164-165 how many record have been used in the statistics? This is no real DMI, the experimental design is confused. What about the experimental units not defined (sometimes cohorts sometimes pens….it is very confused)

L168 with such DMI data to talk of Feed efficiency is very vague

L168-198 with such data quality and two body weights for estimate all the other figures/varibales it is difficult to say that the methodological approach was adequate.

L200-207 assuming that pen=group, the input data are only for a part of the animal: is it correct

Table 2 combining material and method and data here displayed the gain of Galle heifer should be of >70Kg but data reported are different please explain ? too much confusion

Figure 1 data are present using cohorts, why? Big umbalance and I have lost the way: where are pens, etc….nogt coherent with statistcs

Figure 2 is this in line (grey bars) with the access to the system

L297-299 starting form thi “conclusion the authors consided that probably in one farm animals were underfed.

L327 -328 what does it mean ?

Section Calculated growth and methane efficiency traits not clear. How is it possible to get mean 0 with SD e CV ….. the data set is not convinving

Reviewer 2 Report

The objective of this study was to estimate the correlation between feed efficiency and methane emission in beef cattle using large number of animals and studied for linger period. There are many studies on these aspects. But still more studies are required to confirm the methane production, feed efficiency, residual feed intake, residual methane production, etc for genetic selection of animals for breeding purposes with an aim to reduce methane production and increased feed efficiency. The manuscript is usually well prepared. The abbreviations are not used properly throughout the manuscript. Discussion section is lengthy. Please find the comments below.

Other comments

L28: " measured with the GreenFeed" should be deleted here as you did not compare different methane measurement system.

L35: methane or CO2 emission rate should be defined. Is it g/day?

L41-42: Not clear from the abstract how CO2 production could be replaced by DMI. This is your main conclusion, therefore, it should have more support in the abstract.

L143-144: Not clear here. What do you mean by 5 drops? Please clarify it.

Please state clearly how many days of methane measurement was performed for each animal.

Table 1: Units may also be given here.

L171: Is weight body weight?

L172-177: please describe what is meant by contemporary group?

L184:  carbon dioxide to CO2 as you defined earlier. Please follow the carefully the abbreviations used.

L186: use DMI and CO2.

L186-187: Can be modified as The ratio of ADG to CO2 emission rate (ADG/CO2) was first calculated. 

In the subsequent paragraph, please check for the abbreviations.

L226-234: Although diet quality was low in Borcula, what is the possibility strain difference in ADG in the two farms?

L302 -313:  When evaluating the ....  CH4 = 161 ± 70.5 kg/d. These sentences should be deleted as this is not directly related in this study.

L451-452: These equations may be misleading. Did you look for the collinearity (VIF) between the variables. I expect high VIF for MBW and DMI. Otherwise, delete these equations.

L473-476: Not clear how they can be calculated from the SD.

Overall discussion is llengthy. Authors should try to tighten the text to reduce the discussion so that readers can find find interest in the dicussion section.

Reviewer 3 Report

Review of manuscript number: Animals-620279

Title: Methane and carbon dioxide emission of beef heifers in relation with growth and feed efficiency.

The major objective of the study was to precisely estimate the correlation between feed efficiency (feed quality, digestibility, daily body weight growth and others) and methane, carbon dioxide emission measures of beef heifers.

The research was carried out in two herds of beef heifers: Galle (n = 252) and Borculo (n = 74). Heifers were fed ad libitum a roughage diet: grass silage (cultivated fescue) in Galle and natural meadow hay in Borculo. The heifers in Galle did not receive any concentrate complements while the heifers in Borculo received daily one-kilogram complements (composition: extracted soybean meal, corn and barley grains, and sugar cane molasses).

Many necessary studies have been carried out among others; chemical composition of feed, chemical composition of the concentrate complements, chemical composition of feces collected from the caecum (by near infrared reflectance spectroscopy (NIRS) analysis), organic matter digestibility, average daily gain (ADG), mid-test body weight (BW), dry matter intake (DMI), methane (CH4) and carbon dioxide (CO2) emissions - were measured with the GreenFeed systems (with a non-dispersive infrared analyzer). Between all traits in both herds of heifers, a correlation coefficient was calculated.

The researches carried out by the authors are interesting and even important, because they concern the impact on global warming. These are currently modern research, trendy. However, whether the impact of cattle breeding on global greenhouse gas emissions is not exaggerated?

The research required a lot of commitment from the authors; they were well planned and properly conducted. The results are also interesting.

This research can be considered as innovative, but in my opinion the impact of methane and carbon dioxide emissions by herds of cows on global warming should not be overstated, but such research is needed. There is already research on this subject in the available literature, which is why the innovative value is somewhat limited.

The manuscript is generally written very incomprehensibly. Very extensive, too detailed. For the reader it is very difficult to understand. The writing style depends on the authors, but it must be understandable to other readers, even those not related to the topic of research. Work should be greatly simplified.

Introduction

This part is well written, introduces the reader to the subject and further parts of the work.

The purpose of the research should be clearly defined. Currently, the purpose of the work is unclear.

L75 – 79 “Measuring ………. ad libitum.” It should not be in the purpose of work.

Material and methods

This part of the manuscript should be shortened e.g.

L90 -101 Very detailed description. Maybe one sentence about the actual number of heifers (Galle - 252, Borculo - 74) that participated in the study would be enough.

Results and discussion

This part requires significant simplification and removal of unnecessary information, e.g.

L209 – 216 why was this information given?

L251 – 254 this is unnecessary, even not given in this work.

Currently discussion is written in places such as a literature review, e.g.

L407 – 441 The purpose of this part of the work is not to thoroughly discuss the research of other authors, but to discuss the results of own research and compare them to selected studies of other authors.

Conclusions

Specific conclusions should be provided, which results from own research.

According to the reviewer, the work in its current form should not be published. However, this depends on the Editor-in-chief of Animals.

Reviewer 4 Report

The manuscript "Methane and carbon dioxide emission of beef heifers in relation with growth and feed efficiency" is very interesting either in terms of environment pollution or for animal breeding programs. 

I highly appreciated the comparison between two farms adopting different diets  as well as the large number of observation which increase the reliability  of the results.

The authors used adequate methodologies and very well reported and discussed their results, which allowed to give suggestions in breeding programs. 

In my opinion the paper needs only the following corrections and/or addition, before the publication:

line 124: the fecal samples were taken off directly  from the "cecum"? Or form the rectum? Please clarify. 

I suggest to report in a  table the diets chemical characteristics in order to facilitate the readers.  
